# Topical Application of Deglycating Enzymes as an Alternative Non-Invasive Treatment for Presbyopia

**DOI:** 10.3390/ijms24087343

**Published:** 2023-04-16

**Authors:** Joris R. Delanghe, Jeroen Beeckman, Koen Beerens, Jonas Himpe, Nezahat Bostan, Marijn M. Speeckaert, Margo Notebaert, Manon Huizing, Elisabeth Van Aken

**Affiliations:** 1Department of Diagnostic Sciences, Ghent University, 9000 Ghent, Belgiummargo.notebaert@gmail.com (M.N.); 2Department of Electronics and Information Systems, Ghent University, 9000 Ghent, Belgium; jeroen.beeckman@ugent.be; 3Department of Biotechnology, Faculty of Bioscience Engineering, Ghent University, 9000 Ghent, Belgium; koen.beerens@ugent.be; 4Antwerp Biobank, Antwerp University Hospital, 2650 Antwerp, Belgium; nezahat.bostan@uza.be (N.B.);; 5Department of Internal Medicine, Ghent University, 9000 Ghent, Belgium; marijn.speeckaert@ugent.be; 6Research Foundation-Flanders (FWO), 1090 Brussels, Belgium; 7Department of Head and Skin, Ghent University, 9000 Ghent, Belgium

**Keywords:** ageing, fructosyl-amino acid oxidase, glycation, lens, presbyopia

## Abstract

Presbyopia is an age-related vision disorder that is a global public health problem. Up to 85% of people aged ≥40 years develop presbyopia. In 2015, 1.8 billion people globally had presbyopia. Of those with significant near vision disabilities due to uncorrected presbyopia, 94% live in developing countries. Presbyopia is undercorrected in many countries, with reading glasses available for only 6–45% of patients living in developing countries. The high prevalence of uncorrected presbyopia in these parts of the world is due to the lack of adequate diagnosis and affordable treatment. The formation of advanced glycation end products (AGEs) is a non-enzymatic process known as the Maillard reaction. The accumulation of AGEs in the lens contributes to lens aging (leading to presbyopia and cataract formation). Non-enzymatic lens protein glycation induces the gradual accumulation of AGEs in aging lenses. AGE-reducing compounds may be effective at preventing and treating AGE-related processes. Fructosyl-amino acid oxidase (FAOD) is active on both fructosyl lysine and fructosyl valine. As the crosslinks encountered in presbyopia are mainly non-disulfide bridges, and based on the positive results of deglycating enzymes in cataracts (another disease caused by glycation of lens proteins), we studied the ex vivo effects of topical FAOD treatment on the power of human lenses as a new potential non-invasive treatment for presbyopia. This study demonstrated that topical FAOD treatment resulted in an increase in lens power, which is approximately equivalent to the correction obtained by most reading glasses. The best results were obtained for the newer lenses. Simultaneously, a decrease in lens opacity was observed, which improved lens quality. We also demonstrated that topical FAOD treatment results in a breakdown of AGEs, as evidenced by gel permeation chromatography and a marked reduction in autofluorescence. This study demonstrated the therapeutic potential of topical FAOD treatment in presbyopia.

## 1. Introduction

Presbyopia (gradual loss of the eyes’ ability to focus on nearby objects) is a frequently occurring aging-related eye disorder. Presbyopia is a growing global public health issue [1,2]. Both the prevalence and severity of presbyopia increase with age: in the population aged 40 years or older, up to 85% of people develop presbyopia. By 2015, it was reported that 1.8 billion people globally had developed presbyopia. It has been predicted that the prevalence of presbyopia will peak at approximately 2.1 billion by 2030. It has been estimated that 94% of those with significant near vision disability due to uncorrected presbyopia are living in Third World countries [1]. Recent studies have concluded that presbyopia is undercorrected in many countries, with reading glasses only available for 6 to 45% of patients living in developing countries. In lower-income countries, the high prevalence of uncorrected presbyopia is attributable to a lack of adequate diagnosis of the condition and affordable treatment.

Age is generally regarded as the most important risk factor for excessive exposure to ultraviolet rays and related damage to lens proteins (crystallins) caused by free radicals, which are the principal pathogenic components [3]. Because of their very low protein turnover, crystallins (the structural proteins of the human lens) belong to the longest-lived proteins in humans, and thus are prone to a slow, progressive glycation with age. α-, β-, and γ-crystallins constitute more than 90% of the total lens dry mass lens [4]. During the process of protein glycation, free amino groups slowly react with reducing sugars and carbonyls, yielding a variety of adducts that can then rearrange and interact further, eventually leading to lens protein cross-linking. The formation of advanced glycation end products (AGEs) is generally described as the Maillard reaction, a very complex non-enzymatic series of chemical reactions [5]. In the initial steps of the Maillard reaction, a Schiff base (an imine compound) is formed: a reducing sugar (mostly glucose) reacts with a protein-bound amino group. The labile Schiff bases may result in the formation of highly reactive dicarbonyls (e.g., methylglyoxal) or lead to the formation of stable Amadori compounds. The advanced stages of the Maillard reaction eventually lead to the generation of stable adducts, protein cross-links, and AGEs [4,5,6,7].

According to the Maillard theory, the progressive accumulation of AGEs is a major contributing factor to general aging [4,6,7,8]. AGEs have been identified in the human lens during aging [9]. Non-enzymatic glycation of human lens proteins is a major cause responsible for altering lens protein stability and structure, and for inducing covalent cross-linking, aggregation, and insolubilization of the lens crystallins [10]. During aging, increased concentrations of dicarbonyl compounds (e.g., glyoxal and methylglyoxal) also result in AGE cross-links on α-crystallins, leading to a reduction in chaperone activity, enhanced αβ-crystallin content, and the formation of dense aggregates [4]. AGEs gradually accumulate in the aging lens capsules.

AGE-inhibiting or -disrupting compounds may be effective at preventing and treating AGE-related processes. FN3K has been shown to reverse crystalline glycation in cataracts [11]. This process eventually prevents AGE formation. The bacterial enzyme fructosamine oxidase (EC fructosyl-amino acid oxidase, FAOD; fructosyl-α-L-amino acid: oxygen oxidoreductase (defructosylating)) is an enzyme found in many bacteria and yeasts [12,13]. FAOD catalyzes the oxidation of the C-N bond linking C1 of the fructosyl moiety and the nitrogen of the amino group of fructosyl amino acids. Flavin adenine dinucleotide (FAD) acts as a cofactor. It is active on both fructosyl lysine and fructosyl valine. FAOD-based methods for the detection of glycated proteins are commercially available. However, FAOD cannot react with intact hemoglobin A1c (HbA1c), and thus samples require an initial proteolytic digestion step to liberate glycated amino acids or glycated dipeptides [14]. Consequently, its therapeutic use in AGE-induced conditions in humans and animals has not been considered so far. 

At present, there are a number of treatment options for presbyopia, and bifocal or progressive spectacles have been linked with peripheral blur, impaired perception of depth, and a restriction of the visual field, which, in the elderly, have been associated with an increased risk of falls. Contact lens options may be difficult to maintain due to reduced manual dexterity and the development of age-related dry eye symptoms. Other more invasive corrective surgical methods include interventions that alter the optics of the cornea or a replacement of the original crystalline lens. Non-invasive therapies that are currently being investigated include miotic agents and UNR844 (formerly known as EV06), a lipoic acid choline ester [15]. The latter aims to destroy protein disulfide bridges. 

As the crosslinks encountered in presbyopia are mainly non-disulfide bridges, and based on the positive results of deglycating enzymes in cataracts (another eye disease caused by glycation of lens proteins), we studied the ex vivo effects of topical deglycating enzyme (FAOD) treatment on the power of human lenses as a new potential non-invasive treatment for presbyopia. 

## 2. Results

### 2.1. Ex Vivo FAOD Treatment of Lenses

The focal distance of the treated human lenses was monitored over a 5 h incubation period. Figure 1 depicts the evolution of lens power as a function of incubation time. In view of the differences in refractory indices in air and water, 1 diopter (D) in the air corresponds to ±0.16 D in water. Initially, an absolute lens power of 104.3 ± 9.2 D (air) was calculated. After 2 h, the lens power increased by 0.45 ± 0.35 D (water). After 3 h, the average increase in water lens power (water) was 0.96 ± 0.41 D. The maximum average net increase of 12.67 ± 4.02 D (air) was obtained, corresponding to 2.11 ± 0.67 D in water (*p* < 0.001) (Figure 1A). In contrast, neither the inactive FAOD mutant nor the PBS solution resulted in a significant change in lens power during the 5 h incubation period (Figure 1B). FOAD treatment of human lenses resulted in increased lens power.

The trending improvement in lens power was partly dependent on the pre-incubation power of the lens and the age of the lens: y [improvement in lens power, D (water)) = 14.96 − 0.1628x (age, years; *p* = 0.07)] (Figure 2). When studying the opacity of the treated human lenses, a clear improvement was observed.

Over the investigated wavelength interval (400–600 nm), a significant improvement in absorbance was recorded following topical treatment with FAOD: absorbance improvement: 1.47 ± 0.21 (*p* < 0.0001) (Figure 3A). Maximal effects on lens opacity (absorbance untreated—absorbance treated) were observed in the range of 410–430 nm (Figure 3B).

### 2.2. Autofluorescence Kinetics of FAOD Treatment on Human Lens Suspensions

Figure 4 shows the mean change (%) in autofluorescence (AF) values compared with the baseline levels for human lens suspensions with respect to incubation time at a higher concentration range of FAOD: 125 µg/mL, 62.5 µg/mL, and 31.3 µg/mL. Maximum enzymatic digestion was observed after 3 h of incubation with FAOD. While only small or negligible changes in AF values were observed in the control group, remarkable decreases were observed for the enzymes. Similar kinetics were observed at all concentrations.

### 2.3. Gel Permeation Chromatography

Gel permeation chromatography (Figure 5) revealed a marked reduction in the number of fluorescent AGEs. Furthermore, the molecular mass of the peaks decreased following FAOD treatment (4.6 kDa vs. 4.0 kDa treated).

## 3. Discussion

Ex vivo topical FAOD treatment of AGE-modified lenses results in a significant improvement in lens power. FAOD treatment of human lenses results in increased lens power, which reflects a direct relationship with the mechanical properties of the lens [16]. In a single enzyme treatment session, the observed average improvement in lens power is approximately equivalent to most reading glasses in patients with presbyopia. These effects are more pronounced in newer lenses. This finding can be explained by the softer character of the lens in younger individuals due to less advanced cross-linking, which facilitates the diffusion of the FAOD enzyme into the lens. With aging, the development of a barrier to the transport of molecules within the lens has been reported [17]. In general, the physicochemical behavior of enzyme molecules (such as FAOD, 49 kDa) [17] and FN3K (37 kDa) [18] in the 35–50 kDa range allows for swift diffusion in the anterior and posterior eye chambers so that intraocular targets can easily be reached [19]. Reduction in Maillard-type autofluorescence, glycation-induced aggregation, and covalent cross-linking of lens crystallins can result in a decrease in natural elasticity and an increase in stiffness [20,21]. Normally, the lens capsule together with the cortex, when not under the tension of the zonules, causes the lens to assume a rounded shape [21]. 

In our study, FAOD treatment not only improves lens power, but also results in an improvement in lens opacity (and a reduction in stray light). Maximal effects on lens opacity are observed in the wavelength range of 410–430 nm, which is a typical absorbance wavelength of Maillard products that induces a yellow-brownish coloration of aged lenses. Our observations strengthen our hypothesis that the process of non-enzymatic glycation of proteins such as crystallins can be reversed [11,20]. Several studies have demonstrated that Maillard reactions involving sugars can induce a decrease in the chaperone function of α-crystallins [22,23,24,25]. We hypothesize a potential role for FAOD in the restoration of α-crystallin chaperone activity.

Dozens of AGEs have been reported in vivo in cataractous lenses [26]. Tangled AGE structures create a barrier that could complicate the optimal influx of FAOD within the lens interior. However, we hypothesize that the destruction of cross-linked AGEs by FAOD has the ability to reduce the viscosity of the diffusion medium (the lens), thereby facilitating its transport within it. The absence of any therapeutic effect following the administration of the mutant inactive FAOD enzyme proves that the observed effect of the topical application of FAOD is due to the catalytic activity of the FAOD enzyme.

Gel filtration patterns of urea-soluble pooled human lens fractions reveal the presence of high molecular mass compounds, which break down after the FAOD treatment. In parallel, AF kinetics in human lens suspensions reveal a dose- and time-dependent effect of treatment with the FAOD enzyme. These experiments show that topical FAOD treatment results in a deglycation of macromolecular AGEs and in the destruction of cross-links.

Our preliminary ex vivo data might be promising for the cost-effective pharmaceutical treatment of presbyopia. The observed improvements in lens power exceed the average presbyopia correction for most reading glasses. In contrast with pharmacological presbyopia treatment, which is based on inducing miosis [2], the effects of deglycating enzymes on presbyopia are expected to be long-lasting as the rate of (re)glycation is slow. Because enzymes are characterized by a high turnover rate (the maximal number of molecules of substrate converted to product per active site per unit time), the required therapeutic FAOD concentrations are extremely low. The low enzyme concentration in the eye drops (~2 µmol/L) results in intraocular FAOD concentrations in the nmol/L range, which minimizes the chances of adverse effects. Moreover, the human eye is generally considered an immune-privileged organ [27]. 

Our study has several limitations. First, the experiments in this study are only performed on human material in vitro or ex vivo. Human clinical trials are indispensable to assess the clinical validity of our findings. It can be expected that multiple in vivo treatment rounds will be necessary to obtain the most optimal results. In addition, it might also be that after time in humans, crosslinking of the βγ-crystallins recurs, and treatment should be repeated. Finally, the power of our study is hampered by the small number of intact human eye lenses.

## 4. Materials and Methods

### 4.1. Lens Material

Human lenses were obtained from cadaver eyes (n = 14) rejected for corneal transplantation (Biobank Antwerpen, Antwerp Belgium, ID71030031000). The median age of the donors was 74 years (IQR: 74–84 years). Ex vivo FAOD treatment was performed by using an intact left eye lens. The lens from the enucleated human eye was isolated by dissection by a trained ophthalmologist within 24 h post-mortem and stored at 4 °C in RPMI-1640 medium (Thermo Fisher Scientific, Waltham, MA, USA) until analysis. The experiment was started by removing the RPMI medium and washing the lens with 5 mL PBS solution. Fluorescence measurements were performed at baseline at 15 different locations on the ventral and dorsal surfaces of the lens. Subsequently, the whole lens was deglycated by incubation for 20 h at 37 °C in 2 mL of a solution containing 3.84 U/mL FAOD in 0.1 mol/L PBS. After incubation, the lens was washed five times with PBS, and fluorescence measurements were repeated. The lenses were incubated with FAOD, mutant enzymes, or PBS.

### 4.2. Lens Power Measurement

The lens power was measured before treatment and every consecutive hour for up to 4 h. The focal distance measurement is based on the thin lens formula
1/f = 1/S1 + 1/S2
which relates the focal distance of the lens f with the object distance from lens S1 and the image distance from lens S2. An object appears in focus when the three quantities satisfy the thin lens formula. On the other hand, the magnification M of an optical system can be expressed as
M = S2/S1.

By measuring the total distance between the object and image plane
L = S1 + S2
and the magnification M, the focal distance of the lens-under-test can be easily obtained using the equation
L/f = (1 + M)(1 + 1/M).

Because the focal distance of the lens under test is quite small (approximately 8 mm), it is impractical to image directly onto a CMOS sensor, and an extra imaging lens is used. A high-resolution CMOS sensor was used (Basler ace—acA2000-165uc, 2040 × 1086 pixels and pixel size 5.5 µm) in combination with a fixed focal length imaging lens (25 mm/F1.4 59871 Edmund optics, Gloucester Pike, Barrington, NJ, USA). The object consisted of a periodic pattern of dark and white lines. First, a reference image was taken without the lens-under-test by placing the camera in the position where the image was in focus. Then, the lens under test was placed at a random position from the object, and the distance of the camera was repositioned such that the image of the object was in focus. By measuring the period of the lines imaged by the camera, the magnification could be calculated and with the position of the camera, finally, the focal distance could be calculated. The measurement method was verified by performing the procedure for a known aspheric lens with a focal distance of 4.03 mm (Thorlabs C340TMD-A, Newton, NJ, USA), resulting in an error of less than 1% on the focal length. As the quality of the lens-under-test (due to haze and aberrations) was considerably worse than a glass lens, possible issues with focusing the image could be dealt with by repeating the procedure for different distances between the object and lens-under-test and averaging the result.

The relaxed human eye has an approximate optical power of 60 D, with a corneal power of approximately 40 D, or two-thirds of the total power [28]. The crystalline lens of the eye was composed of crystallins with a refractive index of n = 1.40–1.42. In the relaxed eye, the lens had a power of about 20 D, while in the fully accommodated state, it could temporarily increase to 33 D. This means that the lens in the air had a refractive power about 5.4 times stronger than that of the lens located in the eye.

The lens power was measured in air and converted to the lens power in water (the refractive index of the lens in air is 1.5, and the refractive index of the lens in water is 1.13). Figure 6 depicts the method used to assess the lens power in untreated and treated human lenses.

### 4.3. Lens Turbidity

To further assess the optical quality of the human lenses, the opacity of the prepared lenses was measured spectrophotometrically in the wavelength range of 400–600 nm. Measurements were performed using a Cary 60 UV−VIS Spectrophotometer (Agilent Technologies, Santa Clara, CA, USA). To counter the defocusing of the beam path by the lens-under-test, an aspheric glass lens (C060TMD—Thorlabs, Newton, NJ, USA) with a similar focal length was inserted in the beam path. The distance between the two lenses was adjusted to match the original beam size onto the detector so that effectively a beam expander configuration was obtained with a 1× magnification. Absorbance values were recorded in the presence or absence of FAOD treatment.

### 4.4. Gel Filtration

Human lens fragments were obtained after phacoemulsification during cataract surgery in 10 patients. Patients with and without diabetes were included in this study. After surgery, the conservation fluid containing the lens fragments was centrifuged (1900× *g*, 5 min, 21 °C) and the supernatant was removed. Pooled fragments (20 mg) were then incubated for 3 h at 37 °C in 200 µL of a solution containing FAOD (3.83 U/mL). The untreated pooled fragments (20 mg) were stored under the same conditions. As it is well known that cataracts are associated with strong increases in water-insoluble proteins, urea-soluble (water-insoluble) proteins from both untreated and treated lens fragments were extracted in 6 mol/L urea in PBS for 4 h at 4 °C in Eppendorf SafeLock Tubes (Hamburg, Germany). After centrifugation (16,000× *g*, 10 min, 21 °C), the supernatant was retained for gel filtration. Gel filtration of untreated and FAOD-treated lens fragments was carried out on a chromatography column (length: 60 cm, diameter: 15 mm) packed with Sephadex G-25^®^ Fine resin (Sigma-Aldrich, Saint Louis, MO, USA) to assess the molecular mass of fructose-containing lens compounds such as AGEs. Following fractionation, the presence of AGEs was first checked based on Maillard-type AF measurements (excitation 365 nm, emission 390–700 nm) using a Flame miniature spectrometer (FLAME-S-VIS-NIR-ES, 350–1000 nm, Ocean Optics, Dunedin, FL, USA) equipped with a high-power LED light source (365 nm, Ocean Optics) and a reflection probe (QR400-7-VIS-BX, Ocean Optics). Autofluorescence peaks for FAOD-treated lens fragments were detected at 520 nm [29]. Next, all individual fractions were photometrically tested using the resorcinol-HCl (Seliwanoff) reaction, a well-known color reaction for ketoses [30]. In this reaction, a 50 µL sample was added to 100 µL resorcinol (9 mM, Sigma-Aldrich) and 1 mL hydrochloric acid (9 mol/L, Sigma-Aldrich). Following incubation in a boiling water bath for 5 min, the developed color was read photometrically in a standard 10 mm cuvette at 488 nm.

### 4.5. FAOD and FAOD Mutant

Recombinant FAOD from Cryptococcus neoformans, (0.45 U/mg protein; catalog number DIA-409) was purchased from Creative Enzymes (Shirley, NY, USA). Purified proteins were aliquoted, snap-frozen in liquid nitrogen, and stored at −80 °C. An E280L mutant of Aspergillus fumigatus FAOX-II (PDB code 3DJE = UniProt ID: P78573) was produced [18], which was proven to be catalytically inactive.

### 4.6. Autofluorescence Measurement of AGEs

As described previously [20,29], AGEs were quantified based on Maillard-type AF measurements (excitation 365 nm, emission 390–700 nm) using a Flame miniature spectrometer (FLAME-S-VIS-NIR-ES, 350–1000 nm, Ocean Optics, Dunedin, FL, USA) equipped with a high-power LED light source (365 nm, Ocean Optics) and a reflection probe (QR400-7-VIS-BX, Ocean Optics). The measurements were averaged over the 128 scans. AF values were calculated by dividing the average light intensity emitted per nm for the 407–677 nm range by the average light intensity per nm over the 342–407 nm range.

### 4.7. Statistical Analysis

Statistical analyses were performed using GraphPad Prism version 8.4.3. (San Diego, CA, USA). Normality of the data was assessed using the Shapiro−Wilk test. Non-normally distributed data are presented as median with interquartile range (IQR), and normally distributed data as mean ± standard deviation (SD). For non-normally distributed data, unpaired differences between two groups were assessed using the Mann−Whitney U test and paired differences were assessed using the Wilcoxon matched-pairs signed rank test. For normally distributed data, pairwise comparisons between more than two groups were performed using repeated measures of one-way analysis of variance (ANOVA). Subsequently, individual comparisons between the two groups were performed using paired *t*-tests. A *p*-value < 0.05 was considered a priori statistically significant.

## 5. Conclusions

Overall, it can be concluded that, based on in vitro experiments, FAOD is able to break cross-links between lens fibers and deglycate lens crytallins. As the FAOD treatment-induced gain in lens power is in the same magnitude as the vast majority of decreased lens power observed in most presbyopia patients, the FAOD enzyme treatment represents a potential additional treatment option (using enzyme containing eye drops) for AGE-related presbyopia. While our preliminary results need to be validated on larger sample sizes and confirmed by human clinical trials, our findings will pave the way for future research.

## 6. Patents

“Treatment of diseases with fructosyl-amino acid oxidase” patent filed at the European Patent Office. Priority date: March 8 2021 EP21161313. 

## Figures and Tables

**Figure 1 ijms-24-07343-f001:**
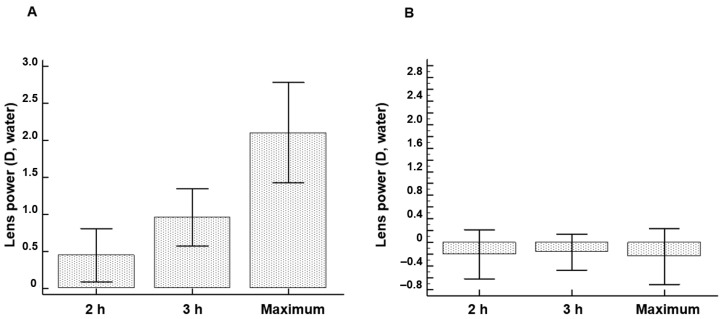
(**A**) Effect of topical FAOD treatment on lens power. (**B**) Effect of PBS solution on lens power.

**Figure 2 ijms-24-07343-f002:**
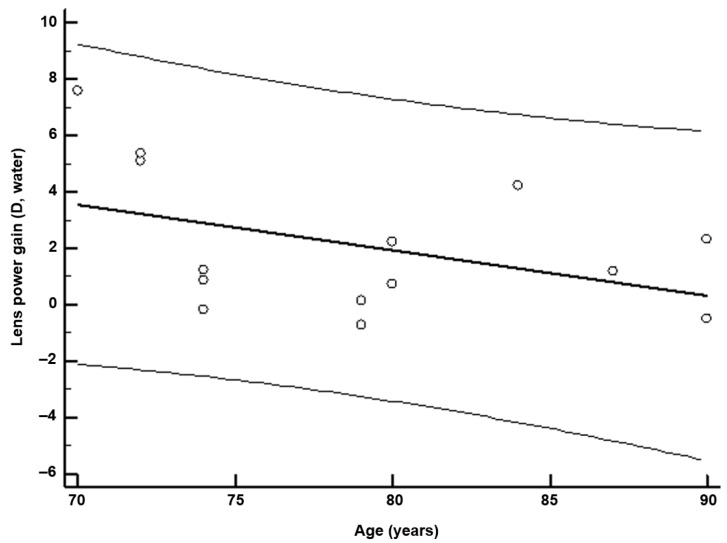
Effect of age on lens power improvement. Y [lens power, D (water)] = 14.96 − 0.163 (age, years) (r^2^ = 0.187, *p* = 0.07).

**Figure 3 ijms-24-07343-f003:**
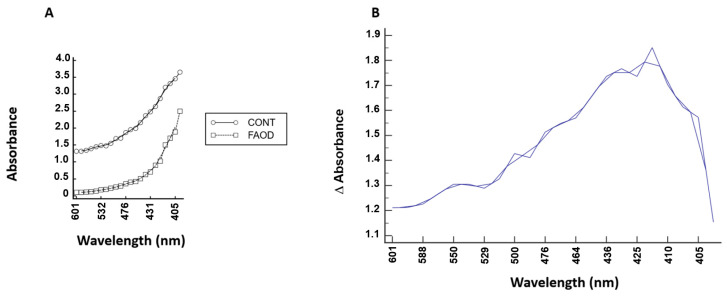
(**A**) Absorbance of a human lens following topical FAOD treatment. Absorbance values are plotted in function of wavelength (FAOD: treated lens; control: untreated lens of the same person). (**B**) Absorbance change in function of wavelength. The net difference in absorbance (absorbance untreated—absorbance treated) is plotted against the wavelength.

**Figure 4 ijms-24-07343-f004:**
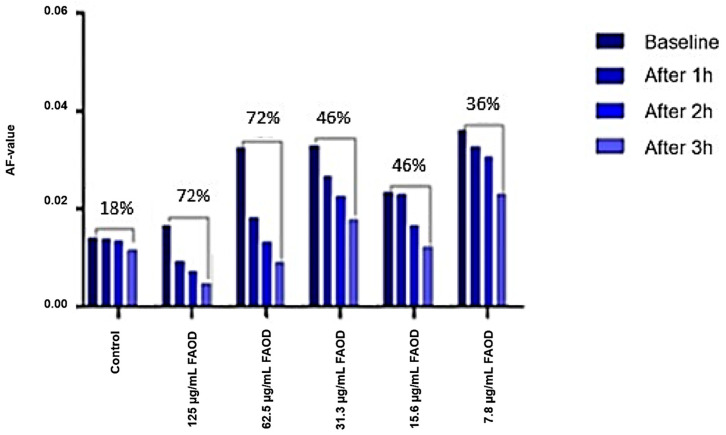
Auto-fluorescence of human lenses following FAOD treatment.

**Figure 5 ijms-24-07343-f005:**
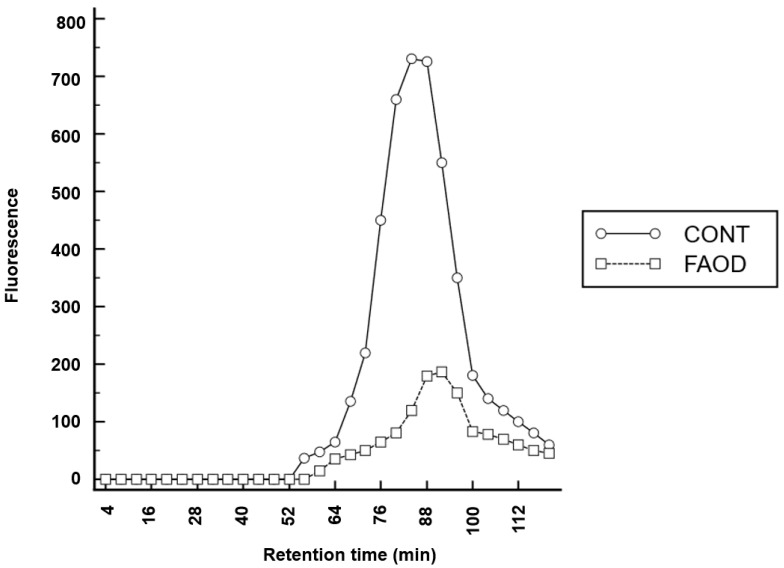
Gel permeation chromatography. Fluorescence (488 nm) following treatment of human lens fragments with the deglycating enzyme FAOD (excitation wavelength 365 nm).

**Figure 6 ijms-24-07343-f006:**
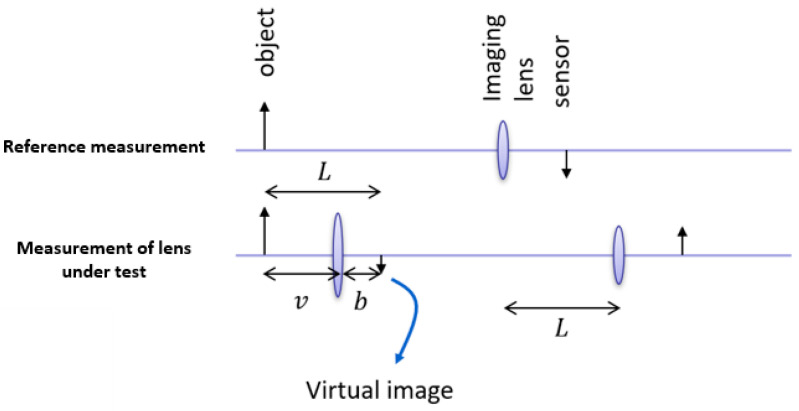
Determination of the focal distance of human lenses.

## Data Availability

The data that support the findings of this study are available from the corresponding author upon reasonable request.

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
