# Peer review of "Topical Application of Deglycating Enzymes as an Alternative Non-Invasive Treatment for Presbyopia"

_ijms, 2023, doi:10.3390/ijms24087343_

Round 1

Reviewer 1 Report

The authors made a very interesting and useful study. Almost everyone over the age of 40 suffers from presbyopia. This phenomenon is based on oxidative damage, which will be even more pronounced in light of the active use of computers and smartphones. Therefore, the incidence of cataract is currently growing. The use of enzymes that allow the destruction of crosslinks has a logical basis. They can be used, probably, not only in diseases of the lens, but also of the cornea. The purpose of the work is clearly defined. Materials and methods are fully consistent with the objectives. The introduction and discussion are well written. However, I have some suggestions for describing the results that will improve the presentation of the study.

1.      Fig. 3a. Please, describe in the caption what is the control.

2.   Fig. 3b. On the y-axis, it is not the absorbance that should be indicated, but the change in the absorbance or delta A.

3. Fig. 4. On the x-axis, please indicate the values of the concentration and not the dilution, and also describe in the caption what is the control.

4. Line 149, replace Figure 5.0 with Figure 5.

5.  In the caption of Fig. 5 specify the fluorescence excitation wavelength.

6.  Figure 5. On the x-axis it is better to indicate not the number of the tube, but the retention time if possible.

7. Line 279. Is the centrifugation rate really 1902g? It seems to me that this is excessive precision and it is enough to write 1900g.

8. The conclusion, in my opinion, looks too short. I think it makes sense to add a few informative sentences to it.

Author Response

REVIEWER 1

The authors made a very interesting and useful study. Almost everyone over the age of 40 suffers from presbyopia. This phenomenon is based on oxidative damage, which will be even more pronounced in light of the active use of computers and smartphones. Therefore, the incidence of cataract is currently growing. The use of enzymes that allow the destruction of crosslinks has a logical basis. They can be used, probably, not only in diseases of the lens, but also of the cornea. The purpose of the work is clearly defined. Materials and methods are fully consistent with the objectives. The introduction and discussion are well written. However, I have some suggestions for describing the results that will improve the presentation of the study.

We thank the reviewer for the constructive comments. We have addressed all remarks as follows:

  1. 3a. Please, describe in the caption what is the control.

This has now been specified

  1. 3b. On the y-axis, it is not the absorbance that should be indicated, but the change in the absorbance or delta A.

This has been corrected in the revised version of the manuscript.

  1. 4. On the x-axis, please indicate the values of the concentration and not the dilution, and also describe in the caption what is the control.

In the revised version of the manuscript, the dilutions have been replaced by the applied FAOD concentrations.

  1. Line 149, replace Figure 5.0 with Figure 5.

This has been done

  1. In the caption of Fig. 5 specify the fluorescence excitation wavelength.

The excitation wavelength (365 nm has been added) to the caption

  1. Figure 5. On the x-axis it is better to indicate not the number of the tube, but the retention time if possible.

This has been changed accordingly.

  1. Line 279. Is the centrifugation rate really 1902g? It seems to me that this is excessive precision and it is enough to write 1900g.

This has been corrected

  1. The conclusion, in my opinion, looks too short. I think it makes sense to add a few informative sentences to it.

The conclusion section has been expanded as follows:

Overall, it can be concluded that, based on in vitro experiments,  FAOD is able to break cross-links between lens fibres and deglycate lens crytallins. As the FAOD tretment-induced gain in lens power is in the same magnitude as the vast majority of decreased lens power observed in most presbyopia patients, the FAOD enzyme treatment represents a potential additional treatment option for AGE-related presbyopia. While our preliminary results need to be validated on larger sample sizes and confirmed by human clinical trials, our findings pave the way for future research.

Reviewer 2 Report

The paper

Topical application of deglycating enzymes as an alternative 2 non-invasive treatment for presbyopia”

demonstrated that ex vivo topical treatment with FAOD can result in the breakdown of AGEs. These results could be promising for the non-invasive and cost-effective treatment of presbyopia.

The work is very interesting and opens new treatment perspectives for a high percentage of people suffering from presbyopia.

For clarity, I would suggest that the equations used be mentioned separately, according to the requirements. It is difficult for the reader to follow.

Author Response

REVIEWER 2

The paper

Topical application of deglycating enzymes as an alternative non-invasive treatment for presbyopia”

demonstrated that ex vivo topical treatment with FAOD can result in the breakdown of AGEs. These results could be promising for the non-invasive and cost-effective treatment of presbyopia.

The work is very interesting and opens new treatment perspectives for a high percentage of people suffering from presbyopia.

For clarity, I would suggest that the equations used be mentioned separately, according to the requirements. It is difficult for the reader to follow.

We thank the reviewer for the constructive comments. We have addressed all remarks as follows:

We have now mentioned the equations separately (and type them in bold), improving the readability of this physics/mathematical paragraph.

The lens power was measured before treatment and every consecutive hour for up to 4 h. The focal distance measurement is based on the thin lens formula

1/f = 1/S1 + 1/S2

which relates the focal distance of the lens f with the object distance from lens S1 and the image distance from lens S2. An object appears in focus when the three quantities satisfy the thin lens formula. On the other hand, the magnification M of an optical system can be expressed as

 M = S2/S1.

By measuring the total distance between the object and image plane

L = S1 + S2

and the magnification M, the focal distance of the lens-under-test can be easily obtained using the equation

L/f = (1+M)(1+1/M).